# Fast Removal of Methylene Blue via Adsorption-Photodegradation on TiO_2_/SBA-15 Synthesized by Slow Calcination

**DOI:** 10.3390/ma15165471

**Published:** 2022-08-09

**Authors:** Maria Ulfa, Hafid Al Afif, Teguh Endah Saraswati, Hasliza Bahruji

**Affiliations:** 1Study Program of Chemistry Education, Faculty of Teacher Training and Education, Sebelas Maret University, Jl. Ir. Sutami 36A, Surakarta 57126, Indonesia; 2Department of Chemistry, Faculty of Mathematics and Natural Sciences, Sebelas Maret University, Jl. Ir. Sutami 36A, Surakarta 57126, Indonesia; 3Centre of Advanced Material and Energy Sciences, University Brunei Darussalam, Jalan Tungku Link, Darussalam BE1410, Brunei

**Keywords:** SBA-15, TiO_2_, slow calcination, low loading, adsorption-photodegradation

## Abstract

TiO_2_/SBA-15 photocatalysts were successfully prepared by impregnating low loading titania to SBA-15 via slow calcination. The photocatalyst is efficient for fast methylene blue removal via adsorption and photodegradation methods. The impregnation of low TiO_2_ loading via slow calcination enhanced TiO_2_ dispersion that preserved the SBA-15 porosity and uniform morphology. High interfacial interaction of TiO_2_/SBA-15 improves TiO_2_ photoresponse by narrowing the bandgap, resulting in a stronger redox ability. The methylene blue removal on 10%TiO_2_/SBA-15 followed the pseudo-second-order kinetic model that reached 67% removal efficiency in 90 min. The synergy between adsorption and photodegradation is responsible for the fast methylene blue removal. These results indicate the importance of maintaining the adsorption capacity in SBA-15 after impregnation with TiO_2_ for efficient adsorption-photodegradation processes, which can be achieved by controlling the deposition of TiO_2_ on SBA-15. A low titania loading further reduced the cost of photocatalysts, thus becoming a potential material for environmental pollution treatment.

## 1. Introduction

The rapidly growing textile industry increased the accumulation of dye pollutants such as methylene blue (C_16_H_18_ClN_3_S), which is harmful to the environment [1]. The threshold value for methylene blue in the water is about 5–10 mg/L, thus requiring the development of an efficient method for removing methylene blue waste [2]. Cheap and effective photocatalysts are continuously being investigated to obtain high efficiency and the fast removal of contamination in water [1,3]. One of the developments in dye treatment is via the combination of adsorption-photodegradation processes [4,5]. Several studies reported using SBA-15 and silica combined with carbon nitride or titania as an efficient photocatalyst for dye removal [4,6,7,8,9]. The method offers simultaneous adsorption of pollutants followed by photodegradation under light irradiation to decompose the adsorbed pollutants into less harmful molecules [6]. These two processes require synergy between the large surface area adsorbent for high accessibility of large dye molecules with the active photocatalysts for degradation [4,10]. TiO_2_ is the most studied photoactive material that can be modified to produce various crystalline phases, structures, and sizes. TiO_2_ is also very active under UV irradiation to catalyze redox reactions [11,12,13,14]. However, TiO_2_ particles are easy to agglomerate, which limits their photodegradation usage. Hence, various methods are employed to increase TiO_2_ stability, such as via the production of nanoparticles and deposition onto high surface area support [15,16].

Deposition of TiO_2_ on Santa Barbara Amorphous-15 (SBA-15) mesoporous silica is a promising way to enhance photocatalytic activity [17,18,19,20]. SBA-15 has a large surface area of 400–800 m^2^/g, large pore diameter with structural regularity, high thermal-physical stability, and thick amorphous silica walls (3–6 nm) [17,21,22,23]. Several studies reported high performances of TiO_2_/SBA-15 [22,24]), TiO_2_/SiO_2_ [25] and TiO_2_/DMS [26] in methylene blue photodegradation. Deposition of TiO_2_ on silica for photocatalytic degradation of Rhodamine B. [27] and methylene blue [22,24] were often produced at 20–60% titania loading [15,19,20,22,24,28]. Additionally, 60–95% efficiencies were obtained at a long removal time of 60–420 min, which involved dark adsorption and photodegradation processes [13,15,22,23,24]. High loading titania and annealing at high temperatures during the synthesis caused the sintering of titania particles, consequently reducing the surface area and blocking the pores of support material.

The slow calcination process improved the decomposition of precursors, facilitated the evacuation of impurities, and produced small particles for high adsorption capacity per unit mass [29,30]. This research investigates the photodegradation of methylene blue via the adsorption-photodegradation process using TiO_2_/SBA-15. A low loading of TiO_2_ at 1.0–10% was deposited on SBA-15 by a slow calcination method to enhance TiO_2_ dispersion and preserve SBA-15 properties. The structural and textural properties of TiO_2_/SBA-15 were determined using XRD, FTIR, Nitrogen adsorption-desorption, and SEM-EDX analysis, followed by the application in methylene blue removal from water.

## 2. Material

Materials used in this study were HCl 37% (Mr 36.5 g/mol), distilled water (Mr 18 g/mol), pluronic triblock copolymer P123 (Mr 5750 g/mol), Tetraethyl orthosilicate (TEOS) (Mr 208.33 g/mol), Tetraethyl orthotitanate (TEOT) (Mr 228.109 g/mol), n-hexane proanalysis (Mr 86.178 g/mol), and Methylene Blue (Mr 319.85 g/mol). All pure chemicals in our analysis were purchased from Sigma Aldrich.

### 2.1. SBA-15 Synthesis

SBA synthesis was performed using the previous study [22] with slight modifications. P123 (4 g) was mixed with HCl solution (19.5 mL of 37% HCl diluted in 127 mL of distilled water) by adding dropwise while stirring at 40 °C, 500 rpm for 3 h. Then, 9.24 mL of TEOS was added to the P123-HCl solution and stirred at the same temperature and speed for 24 h. The resulting solution was put in a polypropylene bottle in an autoclave and heated at 90 °C for 24 h. The resulting solids were filtered, washed, and dried at 100 °C for 24 h, followed by slow calcination. The slow calcination was done by heating the samples from room temperature to ~30 °C at a ramp rate of 1 °C min^−1^, maintaining the temperature for 12 min, then heating to 110 °C for 1 h. Afterward, the heating step was continued to 300 °C for 2 h and increased to 550 °C for 5 h. The next step was slowly cooling them down to room temperature.

### 2.2. Synthesis of TiO_2_/SBA-15

Before impregnation, SBA-15 was activated by mixing 1.5 g of SBA-15 with 50 mL of 0.1 M HCl and then was left for 24 h at room temperature. Afterward, the mixture was filtered, washed several times until the pH was neutral, and dried for 24 h at 100 °C. Impregnation of titania into SBA-15 began with mixing Tetraethyl orthotitanate (TEOT) in 20 mL n-hexane into SBA-15 to obtain 1% TiO_2_/SBA-15 (*w*/*w*). This mixture was stirred at 45 °C for 16 h until a fine powder was formed and then dried at 160 °C for 2 h. The resulting powder was then washed with n-hexane, followed by filtration. The sample was then dried at 80 °C for 45 min. The slow calcination process was then performed on the samples as in the previous procedure to form x-TiO_2_/SBA-15, where x is the amount of TiO_2_ loading in SBA-15.

### 2.3. Characterization

The instruments employed for characterization: X-ray Diffraction (XRD) Pan analytical brand (Version PW3050/60-UK) (Surabaya, Indonesia), Fourier Transform Infrared (FTIR) Spectrophotometer Shimadzu 21 brand with a resolution of 0.5 cm^−1^ (Surakarta, Indonesia), Scanning Electron Microscopy (SEM) with images taken with a JEOL JSM-700 microscope at a voltage speed of 15.0 kV (ITS, Surabaya), and Brunauer-Emmett-Teller (BET) of the Quantacrome Nova 1200 brand (Semarang, Indonesia), as well as a tool for testing samples using a Shimadzu UV-3600 Ultra Violet Visible (UV-Vis) Spectrophotometer with a wavelength of 665 nm (Surakarta, Indonesia). The bandgap energies were obtained using diffuse reflectance ultraviolet-visible (DRUV–Vis) spectroscopy analysis (Malang, Indonesia). The band gap energy (Eg) was determined using a Tauc plot [31,32,33].

### 2.4. Methylene Blue Degradation

The MB removal was performed using a 5 ppm methylene blue solution (200 mL) and 50 mg of 1% TiO_2_/SBA-15 photocatalyst. The mixtures were kept in the dark under stirring and followed with photocatalytic irradiation UV light (125 W Mercury lamp-Philip HPL-N 125W/547 E2, the light intensity 166 mW/cm^2^. The solution was taken every 10 min intervals, and the absorbance of each methylene blue solution was measured using a UV-Vis spectrophotometer at 200–800 nm. Similar steps were repeated for all samples. Analysis of the % efficiency of methylene blue photodegradation using the formula as Equation (1).
(1)$$\%Eff=\frac{Co-Ct}{Co}\times 100\%$$
where Co is the initial concentration of Methylene Blue, *%Eff* is the degradation efficiency, and *Ct* is the concentration of Methylene Blue at t min.

## 3. Result and Discussion

Figure 1 illustrates the wide-angle XRD patterns of SBA-15 and TiO_2_/SBA-15 at different TiO_2_ loading. A wide diffraction peak was detected at around 2θ = 22.5° for SBA-15, indicating the presence of amorphous silica with a stable structure. The broad amorphous silica peak at 22.5° is still visible without significant changes following TiO_2_ deposition. The TiO_2_ peak intensity was enhanced with the increasing TiO_2_ loading. The XRD patterns of SBA-15 after impregnation with 5% and 10% TiO_2_ exhibited small peaks at 2θ = 25.3°, 37.8°, 48.1°, and 65.0° corresponding to the respective (101), (004), (200) (211) anatase crystal planes (JCPDS File No. 21-1286). However, the TiO_2_ peak was absent in 1% TiO_2_/SBA-15 due to the low level of TiO_2_ loading. The small TiO_2_ peaks in 5% and 10% are a typical XRD pattern for low loading of TiO_2_ as the peaks often appeared at loading higher than 12% [22,24,26]. No impurity peak was observed, implying good precursor decomposition during slow calcination.

The morphology and particle size distribution of SBA-15 and TiO_2_/SBA-15 were analyzed using SEM (Figure 2). The SBA-15 existed as a rectangular shape particle with a size estimated at 5.5 µm. After impregnation with 1% titania, the particle size was reduced to 4.3–4.7 µm, with well-defined rectangular shape structures. There is a possibility that TiO_2_ was deposited within the mesopores of SBA-15, preserving the morphological structures of SBA-15. The morphology of TiO_2_/SBA-15 remains unchanged at 5% TiO_2_ loading but slightly loses the well-defined edges following impregnation with 10% TiO_2_. The particle size distribution histogram in Figure 2 showed the decrease of SBA-15 sizes after TiO_2_ impregnation. The slow calcination process during TiO_2_ deposition reduced the agglomeration of silica and titania particles. The deposition of titania inside SBA-15 pores was suggested to occur at low TiO_2_ loading; thus, no changes were observed in the morphology of SBA-15. However, TiO_2_ at 10% loading might be deposited inside the pores and on the outer surface of SBA-15. Nevertheless, the SEM analysis also revealed the stability of the SBA-15 rectangular shape structure even after incorporation with 10% TiO_2_. As shown in XRD, the TiO_2_ peaks observed at 5–10% loading indicate the formation of the crystalline anatase phase of TiO_2_. Slow calcination not only benefits in preserving the structural stability of SBA-15 but also produces a titania crystalline phase as a photoactive site. These results are consistent with the reported studies on the structural stability and high mechanical strength of limestone particles [29] and the reduction of zeolite defects [30] following the slow calcination method.

Figure 3 displays the FTIR spectra of SBA-15 and TiO_2_/SBA-15 at different TiO_2_ loading. The summary of adsorption bands is tabulated in Table 1. The bands at 1,070–1,100 cm^−1^ are visible in all SBA-15 and TiO_2_/SBA-15 samples, which are assigned to the Si-O-Si as asymmetric stretching vibrations mode. The band at 800 cm^−1^ and 460 cm^−1^ assigned to the Si–O–Ti linkage stretching band and Ti-O-Ti bands appeared in all TiO_2_/SBA-15 samples. The intensity of these peaks increased with the titania loading. The Si-O-Ti linkage provides evidence of the successful incorporation of Ti into SiO_2_, which is in line with the previously reported studies [22,24,25,26,34].

Figure 4 presents the nitrogen adsorption isotherm at 77 K for SBA-15 and TiO_2_/SBA-15, while Figure 5 presents their corresponding pore size distributions. The isotherms of these samples are classified as type-IV according to IUPAC, which are typical of mesoporous materials with pore sizes in the nanometer range [21]. The TiO_2_/SBA-15 showed similar isotherm features suggesting the stability of the silica framework in maintaining the mesoporosity despite the addition of TiO_2_. Figure 4 also exhibits hysteresis at a relative pressure (P/P0) of 0.4–0.8, which indicates the presence of mesoporosity. Impregnation of TiO_2_ in SBA-15 does not cause the deformation of the hysteresis loop. However, the surface area and pore volume were reduced with increasing TiO_2_ loading. Figure 5 also showed the pore size distribution at 15–30 Å of pore radius ranges, but with a reduction of pore volume with the increase of titania loading. The results suggest the deposition of TiO_2_ within the pores of SBA-15.

Figure 6 displays the removal efficiency of MB on SBA-15 and TiO_2_/SBA-15 under dark and UV light irradiation. All samples showed removal efficiency at 43–55% under dark adsorption (30 min before UV light irradiation), representing the role of high surface area and mesoporous SBA-15 in removing MB via adsorption. The removal efficiency was further enhanced to reach 59–67% within 60 min of UV irradiation. Table 2 summarizes the results of methylene blue removal using SBA-15 and TiO_2_/SBA-15. In general, TiO_2_/SBA-15 showed a higher removal of methylene blue than SBA-15. Interestingly, MB removal via dark adsorption was enhanced on higher loading TiO_2_/SBA-15 samples. SBA-15 removed 43% of MB via adsorption, with the percentage enhanced to 55% at 10%TiO_2_/SBA-15. There is a possibility that TiO_2_ facilitated the adsorption process by interacting with methylene blue through electrostatic interactions. The role of SBA-15 in absorbing dye pollutants is further increased by the presence of titania. Under UV irradiation, titania removed MB through photodegradation to further added 16.8% of MB removal, reaching the total efficiency of 67.1%. Considering the amount of TiO_2_ used is less than 10%, highly dispersed TiO_2_ particles might be formed on SBA-15, thus exhibiting comparable activities with a large amount of titania, as previously reported in Table 2. Using 10% TiO_2_/SBA-15 showed a total removal efficiency of 67%, almost similar to the reported studies with 21% and 80% TiO_2_ loading [10,22].

The benefits of using a low titania loading are the potential to reduce agglomeration and enhance dispersion, which is strongly important in enhancing the methylene blue photodegradation. The use of low TiO_2_ loading is also beneficial in preserving the adsorption capacity of SBA-15. At 1% TiO_2_ loading, the amount of MB removal is already higher than the reported studies using 46% of TiO_2_ [22] despite having approximately similar surface area. The interaction between adsorbate and adsorbent occurs on SiO_2_ more than TiO_2_ due to the high affinity of the amine group in methylene blue to Si-OH in SBA-15 [28]. Therefore, it is important to provide high surface area support less affected by the pore-blocking effect after titania loading. Figure 7 illustrates the effect of TiO_2_ loading toward agglomeration and dispersion of TiO_2_ that can lead to the pore-blocking effect. The stability of the mesoporous/microporous ratio (Table 2) between 1.97–2.04 regardless of the TiO_2_ loading suggests that using the slow calcination method efficiently dispersed TiO_2_ onto SBA-15 while maintaining the structural properties of SBA-15. Not only low loading of titania but also faster removal of methylene blue than in previous research is explained in Table 2. In general, it can be concluded that the removal capability of methylene blue with TiO_2_/SBA-15 in this study is faster with lower TiO_2_ than in previous studies to save energy and costs in large-scale waste treatment.

The bandgap energy of 10TiO_2_/ZSM-5 was determined using a Tauc plot at 3.05 eV (Figure 8), slightly lower than 3.23 eV of pure TiO_2_ [12]. The reduction of bandgap energy for TiO_2_/SBA-15 catalysts signifies its improved visible-light photoactivity. The narrow bandgap of TiO_2_/SBA-15 allowed the adsorption of photons from a wider wavelength range (A ≤ 400 nm) than TiO_2_ (A ≤ 378 nm)

The mechanism of methylene blue degradation on TiO_2_/SBA-15 photocatalyst under UV light irradiation can be written as follows:
TiO_2_ + hv → TiO_2_ (e^−^ + h^+^)(2)
OH^−^ + h^+^ → OH• (3)
H_2_O + e^−^ → OH• + H^+^(4)
OH• + Methylene blue→ CO_2_ +NO_3_^−^ + SO_4_^2−^ + H_2_O(5)

During light irradiation, TiO_2_/SBA-15 absorbs photon energy for electron excitation to the conduction band, leaving the holes in the valence band (Equation (2)). These photogenerated energy carriers migrate to the surface of TiO_2_/SBA-15 and react with H_2_O or OH- to produce a hydroxyl radical (OH) (Equations (3) and (4)). The hydroxyl radicals are responsible for the redox reaction of the organic pollutants (methylene blue) adsorbed on the surface. The hydroxyl radical is a strong oxidizing agent and acts as the main oxidizer in the photocatalytic oxidation of methylene blue to carbon dioxide, water, and other mineralized products (Equation (5)). The pre-adsorbed methylene blue on the SBA-15 surfaces accelerates the degradation process. The highly dispersed TiO_2_ increases TiO_2_ and SBA-15 interaction to ensure efficient oxidation of the adsorbed methylene blue with the generated hydroxyl radicals.

Figure 9 showed rapid photodegradation of MB in 60 min as determined using the UV spectrophotometer. The rapid reduction of methylene blue absorption is seen in the first 10 min, characterized by a hypsochromic shift ascribed to the change in the chemical structure of the MB. Hydroxyl radicals generated on TiO_2_ caused rapid oxidation of methylene blue, presumably via N-demethylation of dimethylamine groups in MB. The presence of pre-adsorbed MB on the TiO_2_/SBA-15 surfaces further accelerated the degradation process. Overall, about 67% of MB was degraded at about 90 min (consisting of 30 min dark adsorption and 60 min photocatalysis) while using a 10% TiO_2_/SBA-15 photocatalyst. A highly dispersed TiO_2_ with a smaller bandgap and high surface area and porosity of SBA-15 may be responsible for the enhanced activity in MB removal via the adsorption-photodegradation processes.

The kinetic of methylene blue adsorption on SBA-15 and TiO_2_/SBA-15 was evaluated using the pseudo-first-order (Equation (6)) and the pseudo-second-order (Equation (7)) kinetics models using linear regression analysis (Figure 10, Table 3)
(6)$$ln\;\left(qe-qt\right)=ln\;qe-k_{1}t$$
(7)$$t/qt={\frac{1}{k_{2}{\left(qe\right)}^{2}}}^{}-1qe\;t$$
where $k_{1}$ (min^−1^) and $k_{2}$ are the rates of sorption (g mg^−1^min^−1^), qe is the amount of methylene blue removed at equilibrium (mg g^−1^), and qt is the amount of methylene blue removed at t time (mg g^−1^).

Figure 10 and Table 3 showed that all samples followed the pseudo-second model (R^2^ = 0.92–0.95) rather than the pseudo-first-order model (R^2^ = 0.2–0.5). The highest kinetic adsorption-photodegradation rate constant corresponds to 10% titania loading (0.012 min^−1^), suggesting that adsorption accelerates photocatalytic degradation. Adsorption is an important parameter for controlling the kinetic of methylene blue photodegradation. Therefore, generating the synergy between adsorption and photocatalytic degradation is important to ensure the TiO_2_/SBA-15 performances in MB removal. Adsorption can occur via Si-O interaction with the N atom in the methylene blue and hydrogen bonding between OH on silica or titania with the electron pair of N. Table 2 and Table 3 and Figure 9 and Figure 10 present the relationship between the rate of adsorption-photodegradation of methylene blue and the equilibrium point of methylene blue adsorbed on TiO_2_/SBA-15 following variations in titania loading. The removal of methylene blue occurs at a much faster rate with the increased loading of TiO_2_.

## 4. Conclusions

TiO_2_/SBA-15 photocatalysts were obtained using impregnation of low loading titania precursors using slow calcination treatment. SBA-15 was synthesized using TEOS as Si precursor with the addition of P123 as a mesopore template. The studies highlighted the importance of slow calcination by controlling the ramp rate at 1 °C min^−1^ in producing a highly dispersed TiO_2_ while preserving SBA-15 porosity. The TiO_2_/SBA-15 photocatalyst displayed faster adsorption and photodegradation of methylene blue despite having a low TiO_2_ loading. Impregnation with low titania loading on high surface area SBA-15 improves TiO_2_ dispersion for superior photocatalytic activity while simultaneously preserving the surface area and porosity of SBA-15. The pseudo-second-order kinetic model of methylene blue removal established the synergy between adsorption and photocatalytic degradation, which reached total methylene blue removal up to 67%. These results further highlighted the importance of highly dispersed TiO_2_ to maintain the adsorption capacity in SBA-15 for subsequent photodegradation reactions.

## Figures and Tables

**Figure 1 materials-15-05471-f001:**
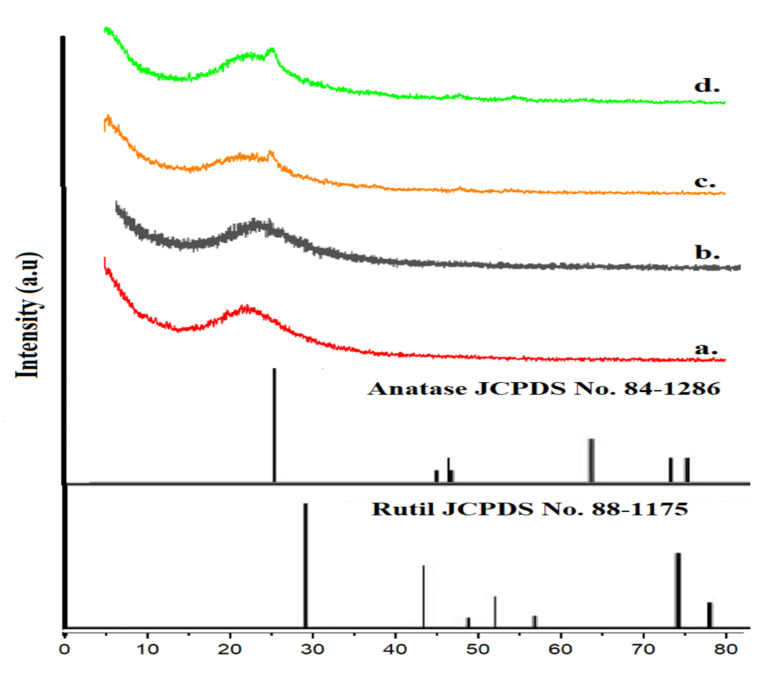
XRD diffractograms of a. SBA-15 and TiO_2_/SBA-15 with loading titania of b. 1%, c. 5%, and d. 10%.

**Figure 2 materials-15-05471-f002:**
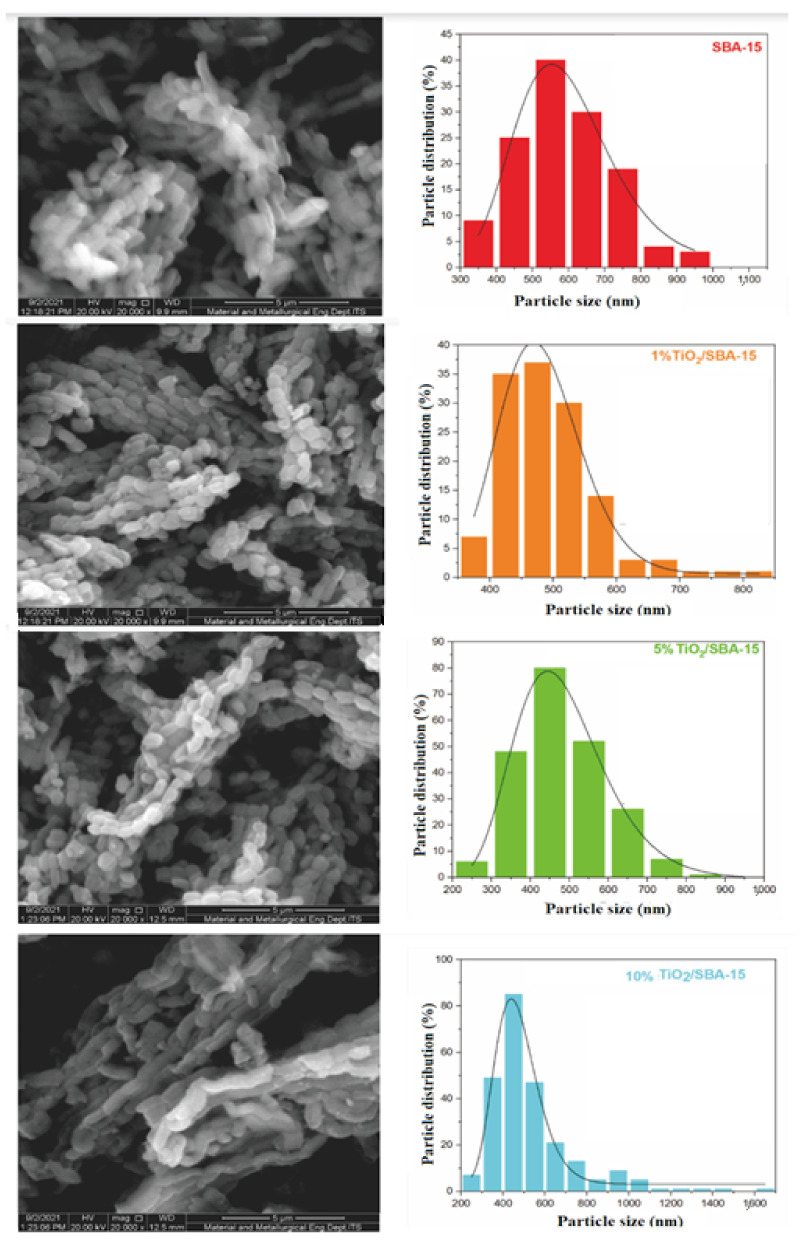
SEM images and particle size distributions of SBA-15 before and after titania impregnation.

**Figure 3 materials-15-05471-f003:**
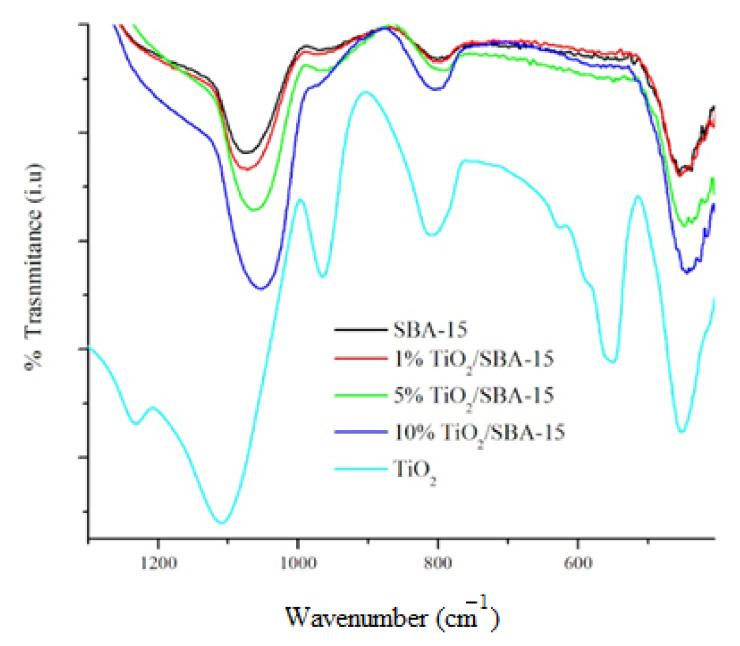
FTIR spectra of SBA-15 and after impregnation with titania.

**Figure 4 materials-15-05471-f004:**
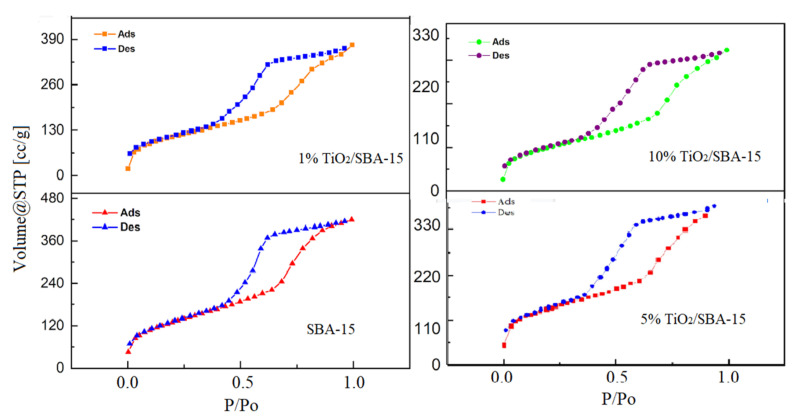
Isotherm nitrogen adsorption-desorption of SBA-15 before and after titania impregnation.

**Figure 5 materials-15-05471-f005:**
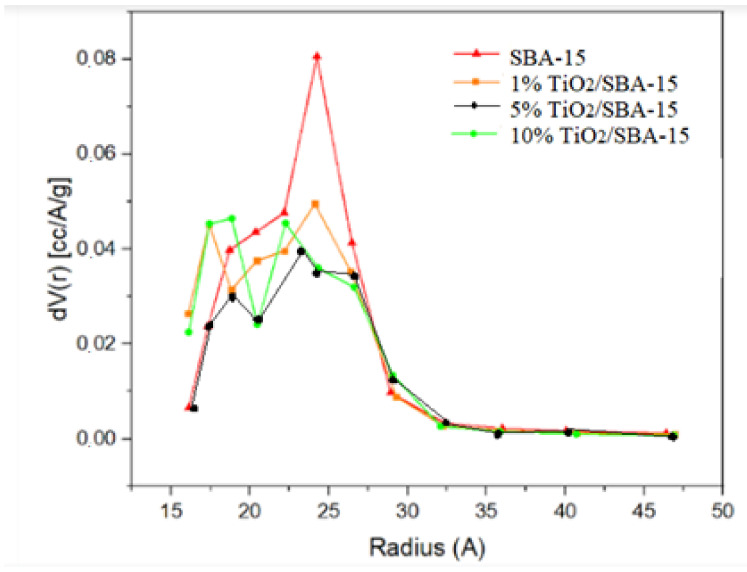
Pore size distribution by BJH method.

**Figure 6 materials-15-05471-f006:**
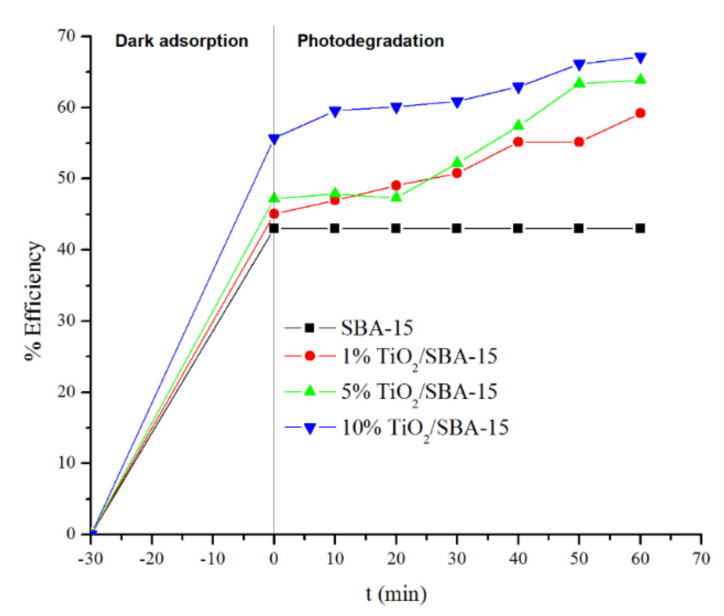
Photodegradation of methylene blue over SBA-15 before and after titania impregnation.

**Figure 7 materials-15-05471-f007:**
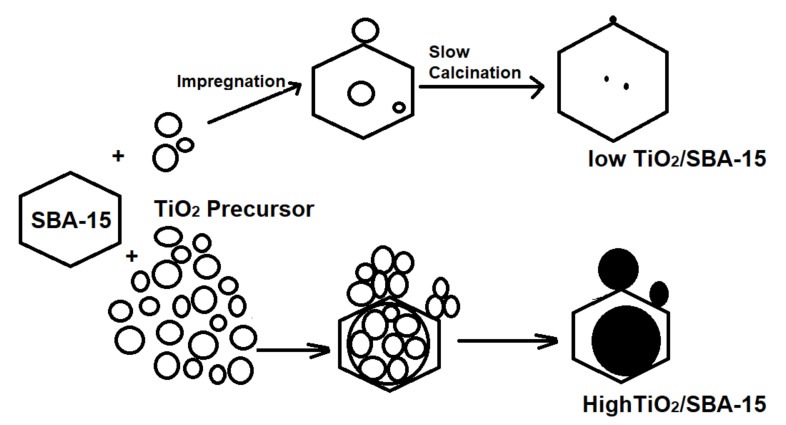
The illustration showed the benefits of the slow calcination process in depositing titania onto SBA-15.

**Figure 8 materials-15-05471-f008:**
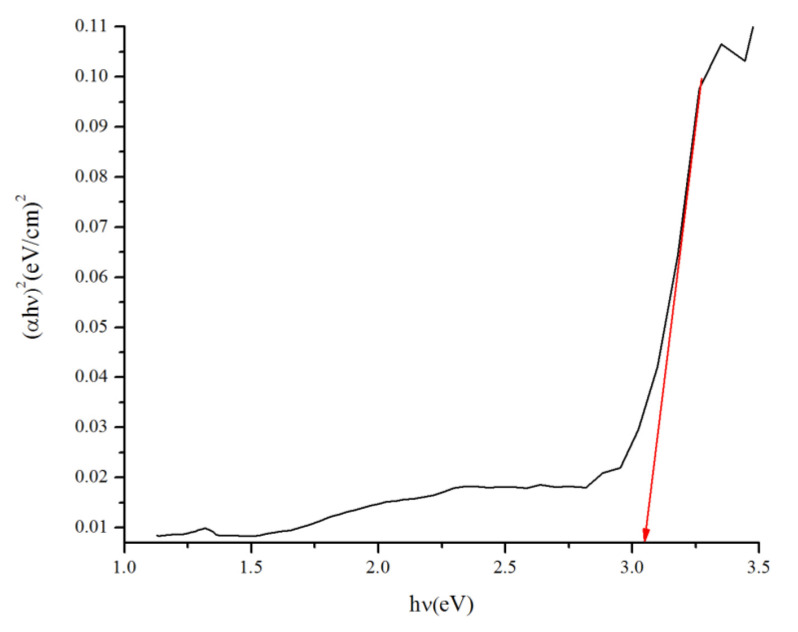
Tauc Plot for 10%TiO_2_/SBA15 samples.

**Figure 9 materials-15-05471-f009:**
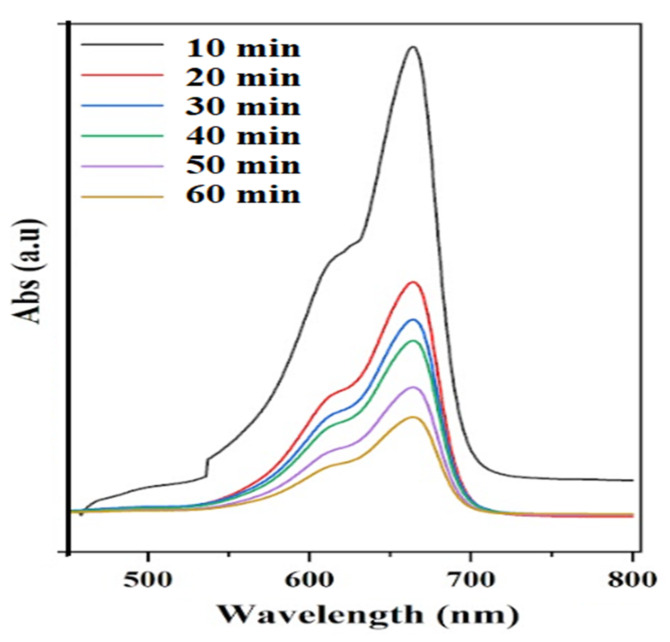
UV-Vis spectra of treated MB solution by photocatalytic degradation using 10% TiO2/SBA-15.

**Figure 10 materials-15-05471-f010:**
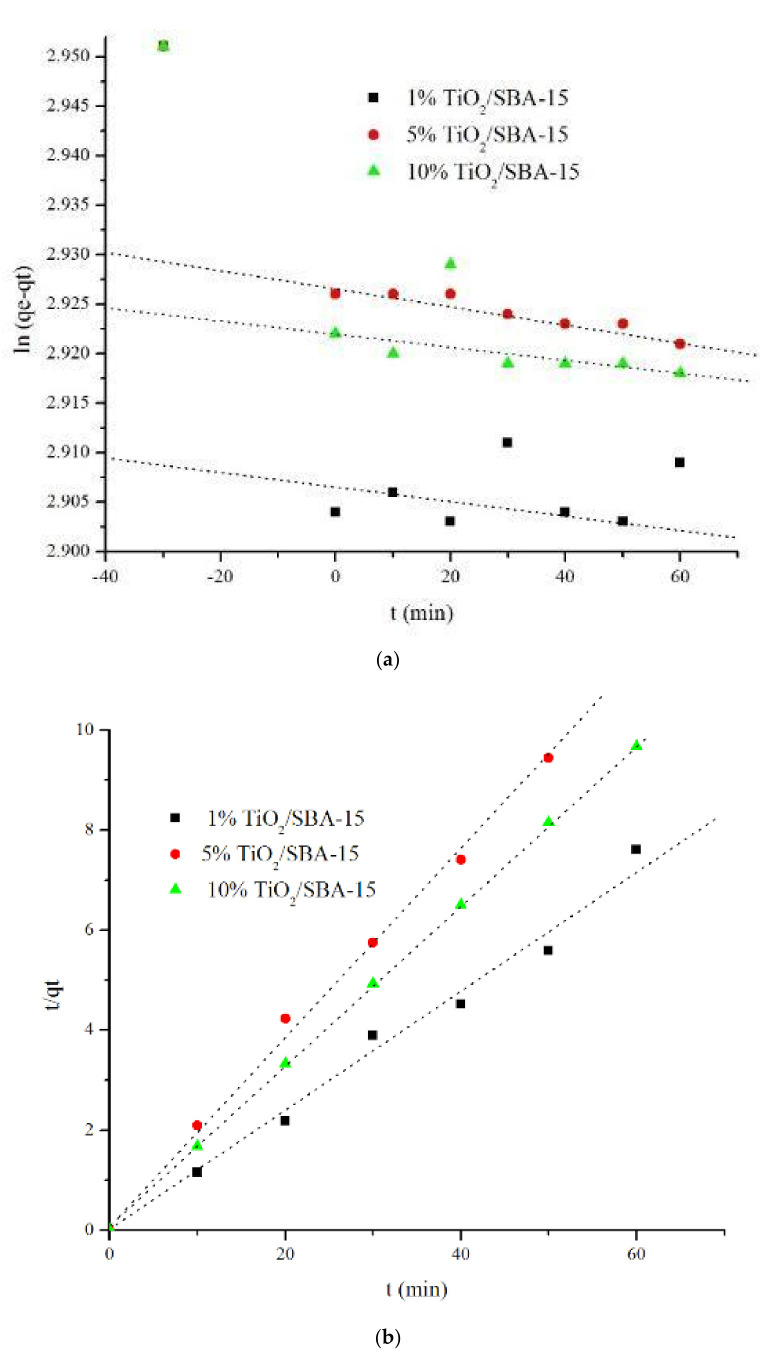
Kinetic models of dark adsorption-photodegradation of methylene blue using TiO_2_/SBA-15 by pseudo (**a**) first and (**b**) second-order.

**Table 1 materials-15-05471-t001:** FTIR analysis data from samples of SBA-15 before and after titania impregnation.

Wave Number (cm^−1^)	Vibration Type [Ref]
SBA-15	
1052–1100	Symmetric stretching vibration Si-O-Si [18]
950–966	Stretching vibration of silanol Si-OH [18]
TiO_2_	
790–800	stretching band Ti-O-Ti [24]
625–550	Ti-O-Ti stretching vibration [12,18]
453	Ti-O-Ti stretching vibration for anatase [12,18]
1%, 5%, 10% TiO/SBA-15	
1052–1100	Symmetric stretching vibration Si-O-Si [18]
940–960	The Si–O–Ti linkage stretching band [24]
790–800	stretching band Ti-O-Ti [24]
440–493	Ti-O-Ti stretching vibration for anatase [12,18]

**Table 2 materials-15-05471-t002:** Efficiency of methylene blue removal by 1, 5, 10 wt% TiO_2_/SBA-15 photocatalyst at 90 min of contact time. The result compares with SBA-15 and reported literature.

Photocatalyst	S^a^ (m^2^/g)	V^b^ (cc/g)	D^c^ (Å)	Vme/Vmi^d^	DA^e^ %	P^f^ %	DA-P^g^ (min)	Total Removal (%)
SBA-15	498	0.737	53.8	2.22	43	0	30–60	43
1-TiO_2_/SBA-15	467	0.648	48.6	2.04	45	14.2	30–60	59.2
5-TiO_2_/SBA-15	399	0.578	48.1	1.99	47	16.8	30–60	63.8
10-TiO_2_/SBA-15	384	0.550	37.4	1.97	55	12.1	30–60	67.1
SBA-15 [22]	753.80	1.430	85.0	-	20	1.5	60–150	21.5
21-TiO_2_/SBA-15 [22]	596	0.88	71.0	-	60	6	60–150	66
80-TiO_2_/SBA-15 [10]	142	0.240	66.5	-	37	3	15–60	40
30-TiO_2_/SBA-15 [10]	499	0.520	41.0	-	65	34	15–60	99
46-TiO_2_/SBA-15 [22]	466	0.52	52.8	-	30	50	60–150	80
SBA-15 [24]	-	-	76	-	2	54	60–420	7
30-TiO_2_/SBA-15 [24]	-	-	65	-	70	28	60–420	98
60-TiO_2_/SBA-15 [24]	-	-	61	-	45	44	60–420	99
SBA-15 [15]	730	1.07	80	-	21	5	120–180	26
29-TiO_2_/SBA-15 [15]	587	0.73	65	-	22	59	120–180	79

S^a^ = Surface area BET, V^b^ = pore volume, D^c^ = pore diameter, Vme/Vmi^d^ = ratio of mesoporous and microporous volume, DA^e^ = % Efficiency from dark adsorption, P^f^ = % Efficiency from Photodegradation, DA-P^g^ = Time of dark adsorption and photodegradation.

**Table 3 materials-15-05471-t003:** Data of kinetics model of Pseudo first order and second order for SBA-15 samples before and after titania impregnation.

Sample	Pseudo First Order		Pseudo Second Order		% Efficiency
	R^2^	k1 (min^−1^)	R^2^	k2 (g mg^−1^min^−1^)	
SBA–15	0.3828	2.2 × 10^−3^	0.950	1.96 × 10^−3^	43.0
1% TiO_2_/SBA-15	0.2346	3.5 × 10^−3^	0.924	1.95 × 10^−3^	59.2
5% TiO_2_/SBA-15	0.6979	4.0 × 10^−3^	0.9558	1.36 × 10^−3^	63.8
10% TiO_2_/SBA-15	0.5344	5.2 × 10^−3^	0.9545	1.29 × 10^−3^	67.1

## Data Availability

Not applicable.

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
