# Peer review of "Fast Removal of Methylene Blue via Adsorption-Photodegradation on TiO2/SBA-15 Synthesized by Slow Calcination"

_materials, 2022, doi:10.3390/ma15165471_

Round 1
Reviewer 1 Report
This is a fairly solid work, which could be probably recommended for publication, but after clarifying some incomprehensible points.
1. Line 27. Reference [1] is not about methylene blue. Reference [3] here is much better.
2. Line 42. Sentence is not correct, because TiO2 is not active center.
3. Lines 43-46. More new information about TiO2. In order for the article to be of interest to a wider readership, more information about titanium oxide would be useful to include. A lot of recent studies have been published in MDPI journals. It is known that TiO2 can be in different crystalline modifications, and the nanostructure is also very diverse, both in structure and size. Few of them are below:
Serga, V.; et al Crystals 2021, 11, 431. https://doi.org/10.3390/cryst11040431
Papachristou, E.; et al. J. Compos. Sci. 2022, 6, 195. https://doi.org/10.3390/jcs6070195
Tsebriienko, T.; et al. Crystals 2021, 11, 794. https://doi.org/10.3390/cryst11070794
4. Fig.1. It is recommended to mark the exact position of the peaks in the figure, otherwise the figure is not clear without a detailed study of the text
5. Fig.2. Was there an evolution of the SEM pictures over time, that is, is it possible to talk about structure stability, or was there a certain aging after all?
6. Fig.3. It is recommended to add an additional table with exact positions of FTIR peaks, especially for region of 400 -1500 cm-1.
7. In general, all link must have year, volume and page number. So, please complete the following references: 4, 8, 13, 16 and 21.
Author Response
"Please see the attachment."

Reviewer 2 Report
As the author mentioned in the introduction, there are many similar researches , and the photocatalysts designed by the author has a poor ability to remove methylene blue, and there is not enough innovation. Furthermore, the authors lack a large number of necessary characterizations, such as electrochemistry, UV-DR, and detailed explanation of the degradation mechanism.
Author Response
"Please see the attachment."

Reviewer 3 Report
The paper is appropriate for publication after major revision concerning:
1) Characterization -line 114, 115. Please explain the significance of "with a wavelength of 665 nm", what represents the 665 nm (peak maximum). Did the authors measure using a single wavelength value?
2) Methylene blue degradation, ...." the process was continued under UV" It should be mentioned the power and spectral range of the lamp used for UV irradiation (Line 120, 121)
3) UV Vis absorption spectra (diffuse reflectance) of the prepared material (catalyst) should be provided
4) UV-visible spectra of MB dye degradation in time in the presence of the catalyst (including the spectral range of interest: 550-750 nm) must be provided.
Author Response
"Please see the attachment."

Round 2
Reviewer 1 Report
The authors have made a successful revision, so the article can be recommended for publication.
Reviewer 3 Report
The paper is appropriate for publication in present form .